# Development of mammary cancer in γ-irradiated F₁ hybrids of susceptible Sprague-Dawley and resistant Copenhagen rats, with copy-number losses that pinpoint potential tumor suppressors

Mayumi Nishimura[1], Kazuhiro Daino[1], Maki Fukuda[2,3¤a], Ikuya Tanaka[2,3¤b], Hitomi Moriyama[1,2¤c], Kaye Showler[2,3¤d], Yukiko Nishimura[1¤e], Masaru Takabatake[1,2], Toshiaki Kokubo[4], Atsuko Ishikawa[1], Kazumasa Inoue[2], Masahiro Fukushi[2], Shizuko Kakinuma[1,2], Tatsuhiko Imaoka[1,2]*, Yoshiya Shimada[1,2¤f]*

1 Department of Radiation Effects Research, National Institute of Radiological Sciences, National Institutes for Quantum and Radiological Science and Technology, Chiba, Japan, 2 Department of Radiological Sciences, Tokyo Metropolitan University, Tokyo, Japan, 3 Radiobiology for Children's Health Research Group, Research Center for Radiation Protection, National Institute of Radiological Sciences, Chiba, Japan, 4 Laboratory Animal and Genome Sciences Section, National Institute of Radiological Sciences, National Institutes for Quantum and Radiological Science and Technology, Chiba, Japan

¤a Current address: Radiology Department, St. Luke's International Hospital, Tokyo, Japan
¤b Current address: Diagnostic Imaging Center, Cancer Institute Hospital of Japanese Foundation for Cancer Research, Tokyo, Japan
¤c Current address: Healthcare Business Headquarters, Konica Minolta Inc., Tokyo, Japan
¤d Current address: Department of Radiology, The Jikei University Hospital, Tokyo, Japan
¤e Current address: Department of Radiobiology, Institute for Environmental Sciences, Aomori, Japan
¤f Current address: Institute for Environmental Sciences, Aomori, Japan
* imaoka.tatsuhiko@qst.go.jp (TI); shimada.yoshiya@ies.or.jp (YS)

**Data Availability Statement:** All data are available from the Gene Expression Omnibus database

## Abstract

Copenhagen rats are highly resistant to mammary carcinogenesis, even after treatment with chemical carcinogens and hormones; most studies indicate that this is a dominant genetic trait. To test whether this trait is also dominant after radiation exposure, we characterized the susceptibility of irradiated Copenhagen rats to mammary carcinogenesis, as well as its inheritance, and identified tumor-suppressor genes that, when inactivated or mutated, may contribute to carcinogenesis. To this end, mammary cancer–susceptible Sprague-Dawley rats, resistant Copenhagen rats, and their F₁ hybrids were irradiated with 4 Gy of γ-rays, and tumor development was monitored. Copy-number variations and allelic imbalances of genomic DNA were studied using microarrays and PCR analysis of polymorphic markers. Gene expression was assessed by quantitative PCR in normal tissues and induced mammary cancers of F₁ rats. Irradiated Copenhagen rats exhibited a very low incidence of mammary cancer. Unexpectedly, this resistance trait did not show dominant inheritance in F₁ rats; rather, they exhibited intermediate susceptibility levels (i.e., between those of their parent strains). The susceptibility of irradiated F₁ rats to the development of benign mammary tumors (i.e., fibroadenoma and adenoma) was also intermediate. Copy-number losses were frequently observed in chromosome regions 1q52–54 (24%), 2q12–15 (33%),

(accession number GSE160514) or are within the manuscript and its Supporting information files.

**Funding:** Japan Society for the Promotion of Science JP17K00562 Kazuhiro Daino Japan Society for the Promotion of Science JP20K12176 Kazuhiro Daino Japan Society for the Promotion of Science JP21H03601 Tatsuhiko Imaoka.

**Competing interests:** The authors have declared that no competing interests exist.

and 3q31–42 (24%), as were focal (38%) and whole (29%) losses of chromosome 5. Some of these chromosomal regions exhibited allelic imbalances. Many cancer-related genes within these regions were downregulated in mammary tumors as compared with normal mammary tissue. Some of the chromosomal losses identified have not been reported previously in chemically induced models, implying a novel mechanism inherent to the irradiated model. Based on these findings, Sprague-Dawley × Copenhagen F$_1$ rats offer a useful model for exploring genes responsible for radiation-induced mammary cancer, which apparently are mainly located in specific regions of chromosomes 1, 2, 3 and 5.

## Introduction

Exposure to ionizing radiation is common in the modern world and can induce various types of DNA damage, including double-strand breaks. Some DNA double strand break repair systems are inherently error-prone; therefore, radiation exposure can result in mutations, such as large deletions, translocations, and reversions, and can ultimately disrupt the integrity and/or expression of cancer-related genes [1]. Thus, radiation is a risk factor for cancer development in humans. Epidemiological studies have been conducted on populations exposed to radiation from various sources, including atomic bombs, medical devices, nuclear industry workplaces, contaminated environments, and natural background (see [2] for an example). These studies have clarified that cancer risk increases with radiation dose in a manner compatible with a linear response, without a threshold, at low doses and low dose rates [2]. They also suggest that the response can be modified by individual factors such as age, sex, and lifestyle factors, including cigarette smoking [3]. Genetic variation is another contributor that governs individual susceptibility to cancer. Familial cancer exhibits a high probability of inheritance and constitutes approximately 5% of all cancers [4], and greater numbers of genetic polymorphisms are considered to influence cancer risk in a more subtle manner [5]. Although some genetic factors that influence the risk of developing acute tissue reactions after high doses of radiation have been identified, little is known about genetic factors that interact with radiation-induced cancer [3]. Clarification of such interaction would be of benefit when considering radiation use in clinical settings, as well as for the selection of emergency workers and astronauts who will be exposed to relatively high doses of radiation [3].

The interaction between genetic and environmental factors often affects the risk of cancer development [6]. As extremely large sample sizes are generally required to identify such interactions in human populations [7], good animal models provide a valuable opportunity for their identification. The rat has been widely used for experimental models of breast cancer because its disease mimics the luminal nature and ductal origin of human breast cancer [8]. Moreover, the characteristics of different rat strains offer opportunities to study breast-cancer resistance/susceptibility and their inheritance. The inbred Copenhagen (COP) rat is almost completely resistant to spontaneous, chemically induced, and hormonally induced mammary carcinogenesis [9–12], but the susceptibility of COP rats to mammary carcinogenesis caused by ionizing radiation (a well-known human breast carcinogen [13]) has not been thoroughly investigated.

The resistance of COP rats to chemically induced mammary carcinogenesis exhibits dominant inheritance when animals are bred with a variety of susceptible strains including Sprague-Dawley (SD) and Wistar-Furth (WF) [11]. The SD rat is probably the most widely used model of breast cancer in the long history of radiation biology research because of its

worldwide availability and high susceptibility to radiation-induced mammary carcinogenesis [14–17]. (SD×COP)$F_1$ rats thus provide a good opportunity to investigate the inheritance of susceptibility to radiation-induced mammary carcinogenesis.

Genetic susceptibility to cancer and environmental factors are often linked to specific types of somatic mutations. For example, non-random genomic changes in chromosome 1 are observed in chemically induced mammary tumors in certain strains, including (WF×COP)$F_1$ rats, but not in radiation-induced cancers [18, 19]. As a potent tumor-suppressor (or oncogenic) gene(s) is expected to be disrupted (or enhanced) in induced cancers in resistant (or susceptible) strains, non-random changes in tumors strongly suggest the existence of potent tumor-related gene(s). Thus, hybrids between strains with different tumor susceptibility offer ideal experimental tools for identifying potentially causative mutations. A successful example is the $F_1$ hybrid of susceptible C57BL/6 and resistant C3H mice, which develops radiation-induced thymic lymphoma with genomic changes in chromosomal regions spanning critical tumor-suppressor genes [20, 21].

In the present study, we investigated the susceptibility of irradiated COP rats to mammary cancer and compared the results with those acquired with SD and (SD×COP)$F_1$ rats to obtain insights into the mode of inheritance. Taking advantage of the mammary carcinomas developed in the (SD×COP)$F_1$ rats, we then located chromosomal regions exhibiting aberrations with the aim of identifying candidate tumor-suppressor genes associated with carcinogenesis.

## Materials and methods

### Animal experiments

Animal experiments were approved by the Institutional Animal Care and Use Committee of the National Institute of Radiological Sciences (NIRS, approval number 07–1014). Female SD rats (Jcl:SD) were purchased from Clea Japan (Tokyo, Japan). COP rats (COP/Hsd) were obtained from Harlan Sprague Dawley (Madison, WI, USA) and maintained by brother-sister mating. $F_1$ hybrid rats were created by crossing female SD and male COP rats at NIRS. Rats were maintained under specific pathogen-free conditions and fed a standard CE-2 diet (Clea Japan) and sterile water ad libitum. Rats of the SD, (SD×COP)$F_1$, and COP strains, born between October 2006 and August 2008, were sequentially irradiated (November 2006–March 2007, May 2007, and July 2007–October 2008, respectively) and observed for overlapping periods of time from November 2006 to September 2010. Experiments were performed as described in detail previously [22]. Briefly, 7-week-old female rats were subjected to whole-body γ-irradiation ($^{137}$Cs, 4 Gy, 0.5 Gy/min) and then palpated weekly for the remainder of their lifetime to detect tumors. The dose and dose rate of radiation and the age at irradiation were chosen as they are established to induce maximal mammary carcinogenesis, based on our previous studies [17, 23, 24]. Animals that showed signs of general deterioration (including signs of natural death) were euthanized by exsanguination under isoflurane anesthesia and autopsied; animals found dead were also autopsied. Animals terminated before tumor development were censored. During autopsy, palpable and non-palpable tumors were collected, fixed in 10% formalin, embedded in paraffin, and processed for hematoxylin and eosin staining for histology [25]. The palpation record was used to determine the age at which tumors first developed. Normal mammary tissue was collected from an approximately 1 cm$^2$ region proximal to an abdominal nipple to ensure inclusion of mammary epithelium; special care was taken to exclude lymph nodes, skin, and muscle. Remaining portions of mammary carcinomas and normal glands were frozen in liquid nitrogen and stored at −80°C. Results of the analysis of certain molecular characteristics of carcinomas for the (SD×COP)$F_1$ cohort were reported

previously [22], as were results pertaining to the development of carcinomas in the SD cohort [17].

### DNA and RNA preparation

DNA extraction from all frozen tumors that were diagnosed as carcinomas ($n$ = 21) was prioritized; of these 21 tumors, those that were freshly collected and had an available remaining portion ($n$ = 10) were used for RNA extraction. This selection of tumors tended to be larger, but were not biased in terms of location, age at detection, age at collection, or the time interval between detection and collection, relative to the entire set of carcinomas ($n$ = 36) (see S1 Fig). Genomic DNA and total RNA were extracted from the same set of frozen normal mammary glands and mammary carcinoma tissue samples using the Maxwell 16 Instrument and System (Promega, Madison, WI, USA) and used for AI and gene expression analyses. Genomic DNA for microarray analysis was extracted as described previously [26].

### Array-based comparative genomic hybridization

DNA (1.25 μg) samples from normal ear and mammary carcinoma samples were labeled with cyanine 3- and cyanine 5-dUTP, respectively, and purified using columns (Agilent Technologies, Santa Clara, CA, USA). Labeled DNA was hybridized with microarray probes (Rat CGH, 2×105K; Agilent) at 65˚C with rotation at 20 rpm for 40 h, and then washed with Wash Buffers 1 and 2 (Agilent). The microarray resolution was ~14.5 kb (on average) with 97,973 probes, which were annotated in the rn4 version of assembly. Microarrays were scanned using the Agilent G2565BA microarray scanner. Fluorescence intensity values were obtained from scanned images with Agilent Feature Extraction software (ver. 9.5.1, Agilent) and were analyzed using DNA Analytics software (ver. 4.0.81, Agilent). Rat orthologs of human genes relevant to breast cancer [27] were identified in the rn4 rat genome assembly using the UCSC Genome Browser (https://genome.ucsc.edu/) [28]. Annotations pertaining to the role of genes in cancer were retrieved from the Oncogene Database (http://ongene.bioinfo-minzhao.org/), Tumor Suppressor Gene Database (https://bioinfo.uth.edu/TSGene/), and COSMIC Database (https://cancer.sanger.ac.uk/census). Microarray data are available at the Gene Expression Omnibus database (https://www.ncbi.nlm.nih.gov/geo/; accession number GSE160514).

### Analysis of allelic imbalance (AI)

AI of a simple sequence length polymorphism genomic DNA locus was assessed by comparing the intensities of two DNA bands amplified with the same PCR primer set. Note that, when intensity of one of the two bands (which represent the two alleles at this locus) is stronger than the other, the imbalance can indicate gain of that allele or loss of the other allele, as the PCR assay used is competitive, rather than quantitative. The primer sequences were obtained from the Rat Genome Database [29]. PCR products were resolved by agarose gel electrophoresis through gels containing ethidium bromide. All AI analyses were performed on genomic DNAs from mammary carcinomas and normal ear skin of the same individual.

### Quantitative PCR

Complementary DNA was synthesized by reverse transcription (ReverTra Ace, Toyobo, Osaka, Japan). Primer sequences and PCR conditions are shown in Table 1 and were validated to amplify a single product of the correct size for each gene by agarose gel electrophoresis. The PCR amplification program consisted of initial denaturation at 95˚C for 10 s followed by 45 amplification cycles of denaturation at 95˚C for 5 s and annealing/elongation at 60˚C for 20 s.

**Table 1. Genes and primers for quantitative PCR.**

| Gene symbol | Chromosome band | Location (Mb) | Gene name | Function | Forward primer (5' → 3') / Reverse primer (5' → 3') |
|---|---|---|---|---|---|
| Asah2 | 1q52 | 236.0 | N-acylsphingosine amidohydrolase 2 | Neutral ceramidase that protects against cytokine-induced apoptosis | GGCATTTGTGAGCGTGGA / TGGGCCAGAGTGAGTGTGA |
| Fas | 1q52 | 238.3 | Fas cell surface death receptor | Receptor that conveys death signal | GAGGGTTTGGAGTTGAAGAGGA / CACGGTTGACAGCAAAATGG |
| Ifit1 | 1q52 | 238.6 | Interferon-induced protein with tetratricopeptide repeats 1 | Involved in cellular response to cytokine stimulus | CCGGAAAGGTGACATAAACGA / AATGTAGGTAGCCAGAGGAAGGTG |
| Sfrp5 | 1q54 | 248.7 | Secreted frizzled-related protein 5 | Involved in several processes including Wnt signaling | GGCCTCATGGAGCAGATGT / CGGTCCCCATTGTCTATCTTG |
| Srek1 | 2q12 | 34.7 | Splicing regulatory glutamic acid and lysine rich protein 1 | Member of family of serine/arginine-rich splicing proteins | GCTGCTTCCCATACCAACCT / AAGTGGTGGCTGTGGTATCTCTC |
| Cenpk | 2q13 | 35.2 | Centromere protein K | Subunit of a centromeric complex | GAAATGTTTGACTGCTGAACTTGG / CCTAATGTTAACAAAACGCCTTCAG |
| Ercc8 | 2q14 | 39.4 | ERCC excision repair 8, CSA ubiquitin ligase complex subunit | Component of nucleotide excision repair | TGGAGTTAAACAAAGACAGGGATG / CTGCTGGCGTTCTCAAGGT |
| Plk2 | 2q14 | 41.8 | Polo like kinase 2 | Serine/threonine protein kinase with role in normal cell division | CCATCATCACCATTCTCACTCC / GATCTGTCATTTCGTAACACTTTGC |
| Gpbp1 | 2q14 | 42.8 | GC-rich promoter binding protein 1 | GC-rich promoter-specific trans-activating transcription factor | AGACACACACATACCCAACCAAA / TGACTGGAGGTTTCCTGCTACTG |
| Il6st | 2q14 | 43.8 | Interleukin 6 signal transducer | Part of cytokine receptor complex | GAAATGTGGTCGGCAAGTCC / ATGGCGGTGTCCATTCTACC |
| Itga1 | 2q14 | 47.2 | Integrin subunit alpha 1 | Subunit of a cell-surface receptor for collagen and laminin | TGGATATTGGCCCTAAGCAGA / TCCCTGTCGGCCTATTTTGT |
| Cat | 3q32 | 88.7 | Catalase | Involved in hydrogen peroxide catabolic process | TGAGAGAGTGGTACATGCAAAGG / GAATCGGACGGCAATAGGAG |
| Meis2 | 3q35 | 101.9 | Meis homeobox 2 | DNA-binding transcription activator in response to growth factor | GTGATTGATGAGAGAGACGGAAG / GCCTGCTGAGTGAGTTGAGG |
| Bmf | 3q35 | 105.0 | Bcl2 modifying factor | Induction of apoptosis | TTGTCCCCTTCTTCCCAATC / ACTGAGGTGGCTCCATGTCTC |
| Rad51 | 3q35 | 105.6 | RAD51 recombinase | Involved in homologous recombination and repair of DNA | GCTGCTTCGACTTGCTGATG / GAGCGATGATGTTTCCTCCAA |
| Tp53bp1 | 3q35 | 108.0 | Tumor protein p53 binding protein 1 | Functions in DNA double-strand break repair pathway choice, promoting 3q36 non-homologous end-joining pathways | TCCGTCAGGCAAAAGGAAAC / CACTCTCACAGGGGCTCACA |
| B2m | 3q35 | 108.9 | β2 microglobulin | Participates in interleukin-12 signaling pathway | CGAGACATGTAATCAAGCTCTATGG / GATGGTGTGCTCATTGCTATTCTT |
| Dusp2 | 3q36 | 114.8 | Dual specificity phosphatase 2 | Phosphatase of mitogen-activated protein kinase, involved in negative regulation | GTTTTGAAAGCTTCCAGGCATACT / GCAAGATTTCCACAGGACCAC |
| Mal | 3q36 | 115.1 | mal, T-cell differentiation protein | Structural constituent of myelin sheath, implicated in metachromatic leukodystrophy | CCTACAGGCATTACCATGAGAACA / CTGGGTTTCAGCTCCCAATC |
| Bcl2l11 | 3q36 | 115.7 | BCL2 like 11 | Interacts with other members of BCL-2 protein family and acts as apoptotic activator | TTACACGAGGAGGGCGTTTG / TCCAGACCAGACGGAAGATG |
| Nbl1 | 3q42 | 148.4 | NBL1, DAN family BMP antagonist | Negative regulation of bone morphogenic protein signaling pathway | TTCCCGCAGTCCACAGAGT / TGCAGTGTACAATCTTCTCAACCA |
| Runx3 | 5q36 | 154.0 | RUNX family transcription factor 3 | DNA-binding transcription factor, implicated in breast cancer | CCTACCACCGAGCCATCAA / AGGCTTTGGTCTGGTCCTCTATC |

(*Continued*)

**Table 1.** (Continued)

| Gene symbol | Chromosome band | Location (Mb) | Gene name | Function | Forward primer (5' → 3') |
|---|---|---|---|---|---|
| | | | | | Reverse primer (5' → 3') |
| *Id3* | 5q36 | 154.9 | Inhibitor of DNA binding 3, HLH protein | Involved in positive regulation of apoptosis | GTGATCTCCAAGGACAAGAGGAG |
| | | | | | TGGAGAGAGGGTCCCAGAGT |
| *C1qa* | 5q36 | 155.7 | Complement C1q A chain | Participates in coagulation cascade | CGGGTCTCAAAGGAGAGAGAGG |
| | | | | | CCCACATTGCCGGGTTT |
| *Gapdh* | 4q42 | 161.3 | Glyceraldehyde-3-phosphate dehydrogenase | Participates in gluconeogenesis pathway, used as an internal control | TCAACGGGAAACCCATCAC |
| | | | | | TTTTGGCCCCACCCTTC |

PCR was performed using the Stratagene Mx3000P real-time PCR system (Agilent) and SYBR Premix Ex Taq (Takara Bio, Otsu, Japan). The expression levels of target genes were normalized to those of *Gapdh* and expressed relative to the value of an arbitrarily selected normal tissue sample using the $2^{-\Delta\Delta Ct}$ method [30].

## Statistics

Tumor incidence was compared using Fisher's exact test. Tumor number and age of first tumor detection were assessed by the Kruskal-Wallis test followed by pairwise comparison using the Mann-Whitney *U* test. Tumor-free survival data were analyzed by the log rank test and Cox regression. Gene expression levels in two groups were compared with the Mann-Whitney *U* test. The significance of correlations was assessed by Spearman's rank correlation test. *P* values < 0.05 were considered statistically significant. Analyses were performed on statistical software R [31].

## Results

### Irradiated COP rats are less susceptible to mammary cancer than irradiated SD or (SD×COP)F$_1$ rats

To understand the susceptibility of irradiated COP and (SD×COP)F$_1$ rats to mammary carcinogenesis, we irradiated each of SD (*n* = 20), COP (*n* = 19), and (SD×COP)F$_1$ (*n* = 29) rats with 4 Gy of γ-rays and monitored the development of palpable mammary carcinomas and benign tumors. Benign tumors consisted of fibroadenoma (~95%) and adenoma (~5%). All carcinomas and benign tumors were palpable, and no additional tumors in these categories were discovered upon necropsy. Neither the location of tumors (abdomino-inguinal or thoracic) nor the malignant-to-benign ratio differed among strains, although COP rats had more adenomas than SD and (SD×COP)F$_1$ rats as benign tumors (Table 2). Despite the significantly shortened observation period for SD rats (Table 3), the percentage of rats having carcinomas and benign tumors during their lifetime was higher for SD rats than COP rats, and SD rats had a greater number of tumors per rat (Table 3). The age at the first palpation of individual tumors was also lower for SD rats (Table 3). (SD×COP)F$_1$ rats showed susceptibility that was close to that of SD rats (Table 3). Analysis of the time to first palpable mammary carcinoma indicated that COP rats developed carcinoma significantly less frequently than SD and (SD×COP)F$_1$ rats, whereas the difference between SD and (SD×COP)F$_1$ rats was also substantial, albeit with only marginal statistical significance, suggesting intermediate susceptibility of the (SD×COP)F$_1$ rats (Fig 1A and Table 4). Susceptibility of (SD×COP)F$_1$ rats to benign mammary tumors was intermediate between the parental strains (Fig 1B and Table 4). Causes of censoring did not differ among strains (Table 5). Throughout the course of the experiment,

**Table 2. Distribution of tumor location and type.**

| Strain | Location (carcinoma) | | Location (benign) | | Tumor type (all) | | Tumor type (benign tumors) | |
|---|---|---|---|---|---|---|---|---|
| | Abdomino-inguinal | Thoracic | Abdomino-inguinal | Thoracic | Malignant [a] | Benign [b] | Fibroadenoma | Adenoma |
| SD | 12 (67%) | 6 (33%) | 27 (51%) | 26 (49%) | 18/71 (25%) | 53/71 (75%) | 53/53 (100%) | 0/53 (0%)*** |
| COP | 8 (89%) | 1 (11%) | 5 (45%) | 6 (55%) | 9/20 (45%) | 11/20 (55%) | 7/11 (64%) | 4/11 (36%) |
| (SD×COP)F$_1$ | 18 (50%) | 18 (50%) | 38 (56%) | 30 (44%) | 36/104 (35%) | 68/104 (65%) | 65/68 (96%) | 3/68 (4%)** |

[a] Carcinoma,

[b] fibroadenoma and adenoma.

**P < 0.01,

***P < 0.001 vs. COP.

SD rats were significantly heavier than COP rats, while the weights of (SD×COP)F$_1$ rats were intermediate (Fig 1C). Thus, irradiated (SD×COP)F$_1$ rats showed marginally higher mammary-cancer susceptibility, in contrast to the reported dominant inheritance of resistance of COP rats to chemically induced mammary carcinogenesis.

## Mammary carcinomas of (SD×COP) F$_1$ rats have multiple localized copy-number variations

Previous studies have shown that radiation-induced mammary carcinomas of SD rats harbor multiple copy-number aberrations that do not converge to specific chromosomal regions [26, 32, 33]. In surprising contrast, our present analysis of mammary carcinomas ($n$ = 21) from (SD×COP)F$_1$ rats revealed multiple copy-number variations in several specific chromosomal regions (Fig 2A). These variations included copy-number losses of chromosome 1q52–54 (observed in 5 carcinomas, 24%), 2q12–15 (7 carcinomas, 33%), and 3q31–42 (5 carcinomas, 24%). Additional large deletions spanning nearly all of chromosome 5 were identified in 6 carcinomas (29%). Focal deletions involving *Cdkn2a* and *Cdkn2b* were found at chromosome 5q32 in 2 carcinomas (10%), as had been repeatedly observed in a subset of radiation-induced rat mammary carcinomas [26, 32, 33]. Combined, focal and large deletions affected this chromosomal region in 8 carcinomas (38%). Deletions of 5q36 were found in 7 carcinomas (33%). Copy-number gains were relatively rare (Fig 2A). We identified many genes relevant to human breast cancer [27] in the chromosomal regions exhibiting copy-number changes

**Table 3. Crude analysis of mammary tumor development in SD, COP and (SD×COP)F$_1$ strains.**

| Strain | Age at autopsy (weeks)[a] | Rats with tumor (%) | | Tumors per rat[b] | | Age at first tumor detection (weeks)[a] | |
|---|---|---|---|---|---|---|---|
| | | Carcinoma | Benign | Carcinoma | Benign | Carcinoma | Benign |
| SD | 64.4 ± 19.2 | 13/20 (65) | 17/20 (85) | 0.90 ± 0.20 | 2.65 ± 0.53 | 50.5 ± 25.9 | 60.1 ± 16.5 |
| COP | 92.8 ± 17.1*** | 7/19 (37) | 10/19 (53) | 0.47 ± 0.16 | 0.58 ± 0.14*** | 93.1 ± 14.1*** | 76.2 ± 18.2* |
| (SD×COP)F$_1$ | 82.0 ± 23.7** | 19/29 (66) | 21/29 (72) | 1.24 ± 0.27† | 2.34 ± 0.36†† | 70.1 ± 23.3*†† | 77.8 ± 18.9*** |

[a]Mean ± standard deviation,

[b]mean ± standard error of the mean.

*P < 0.05,

**P < 0.01,

***P < 0.001 vs. SD;

†P < 0.05,

††P < 0.01 vs. COP.

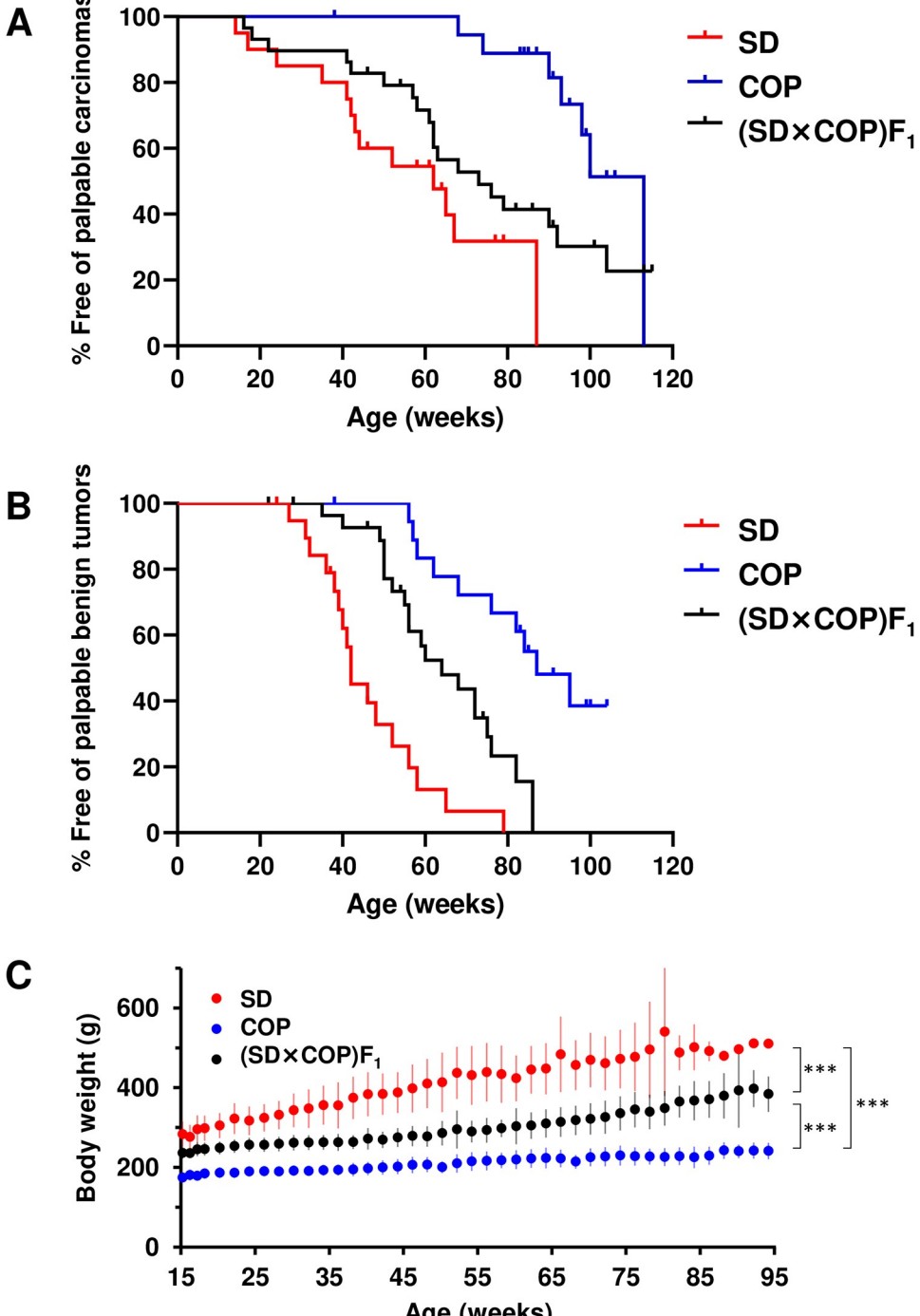

**Fig 1. Kaplan-Meier plots of palpable mammary tumor development in Copenhagen (COP), Sprague-Dawley (SD) and hybrid [(SD×COP)F₁] rats.** A, carcinoma; B, benign tumors (fibroadenoma and adenoma). Data from SD rats were reported previously [17] and reanalyzed. C, Body weight during the experiment (mean and standard deviation). \*\*\* $P < 0.001$ between strains, two-way analysis of variance.

**Table 4. Hazard analysis of palpable mammary tumors among strains.**

| Strain | Hazard ratio (vs. SD) | Log rank test (vs. SD) | Hazard ratio (vs. COP) | Log rank test (vs. COP) |
|---|---|---|---|---|
| *Carcinomas* | | | | |
| SD | 1 (referent) | — | 6.2 (2.3, 17) | $P = 1 \times 10^{-5}$ |
| COP | 0.16 (0.06, 0.43) | $P = 1 \times 10^{-5}$ | 1 (referent) | — |
| (SD×COP)F$_1$ | 0.46 (0.21, 1.0) | $P = 0.1$ | 2.8 (1.2, 6.8) | $P = 0.02$ |
| *Benign tumors* | | | | |
| SD | 1 (referent) | — | 14 (5.5, 36) | $P = 1 \times 10^{-7}$ |
| COP | 0.07 (0.03, 0.18) | $P = 1 \times 10^{-7}$ | 1 (referent) | — |
| (SD×COP)F$_1$ | 0.29 (0.15, 0.57) | $P = 0.0002$ | 4.1 (1.7, 10) | $P = 0.0006$ |

Numbers in parentheses denote 95% confidence interval.

(Table 6). Interestingly, the number of genes affected by copy-number loss was positively correlated with age at tumor detection (Fig 2B, $P < 0.01$), which could be explained by correlation with the number of chromosomes with large deletions spanning >80% of the chromosome (Fig 2C, $P < 0.01$). The frequencies of copy-number losses and gains involving human-relevant tumor suppressors and proto-oncogenes, respectively, were much higher than those reported in a previous study on SD rats [33] (Table 6, two rightmost columns).

## Mammary carcinomas of (SD×COP)F$_1$ rats have multiple AIs

A previous study on radiation-induced mammary carcinoma of (WF×COP)F$_1$ rats indicated that AIs occur at very low frequency (4–13%) in these tumors; the study did not relate these AIs to copy number aberrations [18]. As the SD strain is an outbred population and genetically heterogeneous, we first searched for simple sequence length polymorphism markers that showed heterozygosity in (SD×COP)F$_1$ rats and their parental strains, using genomic DNA obtained from normal ear skin. This search identified 42 markers that showed heterozygosity in tumor-bearing (SD×COP)F$_1$ rats (Fig 3, markers in black letters with asterisks). We found 35 instances of AIs across 20 markers in mammary carcinoma genomes from (SD×COP)F$_{1f}$ rats (Fig 3, COP and SD). The frequently lost regions (Fig 3, light grey) of chromosomes 1, 2, 3 and 5 identified in the microarray analysis coincided with markers D1Rat49, D1Rat67, D1Mgh29 (chromosome 1), D2Rat116, D2Rat17 (chromosome 2), D3Mit7, D3Rat164

**Table 5. Causes of censoring.**

| Analysis | Cause | Strain* | | |
|---|---|---|---|---|
| | | SD | COP | (SD×COP)F$_1$ |
| Carcinoma | Mammary neoplasm | 4 | 2 | 5 |
| | Other neoplasms | 1 | 3 | 3 |
| | Non-neoplasms | 0 | 3 | 0 |
| | Unidentified | 2 | 4 | 2 |
| Benign tumors | Mammary neoplasm | 2 | 1 | 5 |
| | Other neoplasms | 1 | 2 | 1 |
| | Non-neoplasms | 0 | 1 | 0 |
| | Unidentified | 0 | 5 | 2 |

* There was no significant difference among strains (Fisher's exact test).

SD, Sprague-Dawley; COP, Copenhagen.

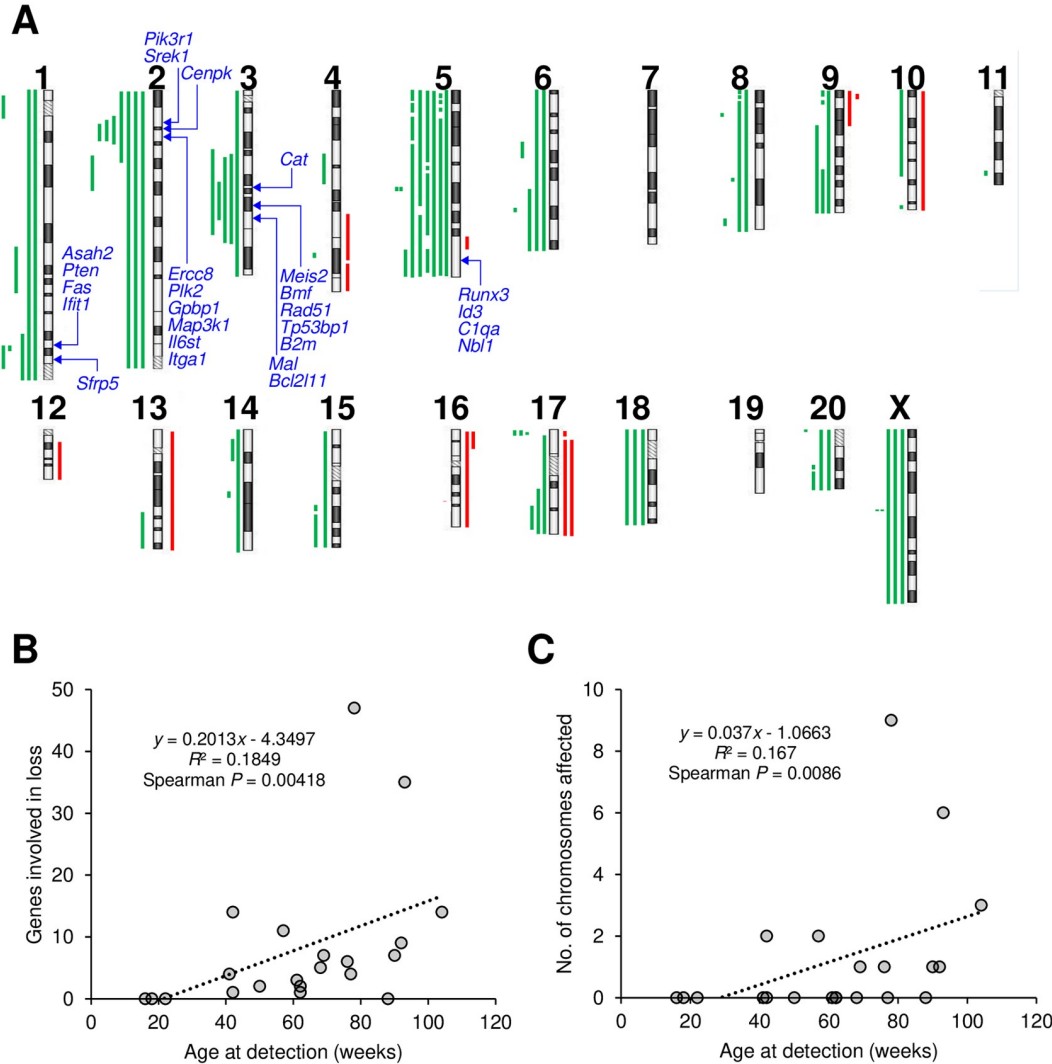

**Fig 2. Copy-number aberrations in mammary carcinomas from (SD×COP)F₁ rats.** A, Chromosomal losses (green) and gains (red) in 21 tumors. Aberrations observed in the same carcinoma are arranged vertically, whereas those in different carcinomas are side by side. Locations of genes examined in the expression analysis are shown in blue (see Fig 4 and text). B and C. Correlation between age at tumor detection and genomic changes in tumors. B, Number of breast cancer–related genes affected by copy number loss. C, Number of chromosomes with >80% copy number loss.

(chromosome 3), and D5Mit14 (chromosome 5). In AIs of these markers, imbalance was not strongly biased towards either the SD or COP allele, and the trend varied among chromosomal sites (Fig 3, SD and COP). At several sites, copy-number variations identified using microarrays did not accompany AI, indicating the relatively low sensitivity of AI detection, which is understandable, considering the possibility for contamination of tumor samples with non-malignant cells (e.g., stromal cells). The majority of observed AIs (28/35) were accompanied by microarray-identified copy-number loss (Fig 3, SD and COP on light-grey background), suggesting that these AIs reflected a loss of heterozygosity produced by large deletions. In contrast, AIs accompanying microarray-identified copy-number gain were rare (2/35; Fig 3, SD and COP on black background). Some AIs (5/35) were without copy-number variation (Fig 3,

**Table 6. Human breast cancer–related genes in chromosomal regions exhibiting meaningful copy-number changes in (SD×COP)F$_1$ mammary carcinoma.**

| Gene symbol | Role in cancer[a] | Chromosome band | Location (Mb) | Tumors with copy-number change (n = 21) | | Moriyama et al. [33] (n = 28)[b] | |
|---|---|---|---|---|---|---|---|
| | | | | Loss | Gain | Loss | Gain |
| Esr1 | POG/TSG | 1q11 | 35.5–35.8 | 2 (10%) | 0 (0%) | 0 (0%) | 0 (0%) |
| Cnot3 | TSG | 1q12 | 63.9 | 2 (10%) | 0 (0%) | 0 (0%) | 0 (0%) |
| Cic | POG/TSG | 1q21 | 80.6 | 2 (10%) | 0 (0%) | 0 (0%) | 1 (4%) |
| Palb2 | TSG | 1q36 | 180.9–181.0 | 3 (14%) | 0 (0%) | 0 (0%) | 0 (0%) |
| Men1 | TSG | 1q43 | 209.1 | 2 (10%) | 0 (0%) | 0 (0%) | 0 (0%) |
| Pten | TSG | 1q52 | 236.8 | 5 (24%) | 0 (0%) | 2 (7%) | 0 (0%) |
| Pik3r1 | POG/TSG | 2q12 | 32.6–32.7 | 7 (33%) | 0 (0%) | 0 (0%) | 0 (0%) |
| Map3k1 | POG/TSG | 2q14 | 43.1 | 7 (33%) | 0 (0%) | 0 (0%) | 0 (0%) |
| Fbxw7 | TSG | 2q34 | 176.7–176.8 | 3 (14%) | 0 (0%) | 0 (0%) | 0 (0%) |
| Notch2 | POG/TSG | 2q34 | 192.8–193.0 | 3 (14%) | 0 (0%) | 0 (0%) | 0 (0%) |
| Notch1 | POG/TSG | 3p13 | 4.6–4.7 | 1 (5%) | 0 (0%) | 0 (0%) | 1 (4%) |
| Bub1b | TSG | 3q35 | 105.1 | 5 (24%) | 0 (0%) | 0 (0%) | 0 (0%) |
| Foxp1 | POG/TSG | 4q34 | 133.8–134.0 | 1 (5%) | 0 (0%) | 0 (0%) | 0 (0%) |
| Cdkn1b | POG/TSG | 4q43 | 171.8 | 1 (5%) | 0 (0%) | 0 (0%) | 0 (0%) |
| Cdkn2a | TSG | 5q32 | 108.9 | 8 (38%) | 0 (0%) | 3 (11%) | 0 (0%) |
| Cdkn2b | TSG | 5q32 | 108.9 | 8 (38%) | 0 (0%) | 3 (11%) | 0 (0%) |
| Arid1a | TSG | 5q36 | 151.4 | 7 (33%) | 0 (0%) | 0 (0%) | 0 (0%) |
| Spen | TSG | 5q36 | 160.4 | 7 (33%) | 0 (0%) | 0 (0%) | 0 (0%) |
| Msh2 | TSG | 6q12 | 11.2–11.3 | 2 (10%) | 0 (0%) | 1 (4%) | 0 (0%) |
| Dnmt3a | TSG | 6q14 | 26.8–26.9 | 2 (10%) | 0 (0%) | 0 (0%) | 0 (0%) |
| Smarca4 | TSG | 8q13 | 20.7–20.8 | 2 (10%) | 0 (0%) | 0 (0%) | 0 (0%) |
| Atm | TSG | 8q24 | 56.9–57.0 | 2 (10%) | 0 (0%) | 0 (0%) | 0 (0%) |
| Setd2 | TSG | 8q32 | 114.9–115 | 2 (10%) | 0 (0%) | 0 (0%) | 0 (0%) |
| Mlh1 | TSG | 8q32 | 115.6–115.7 | 2 (10%) | 0 (0%) | 1 (4%) | 0 (0%) |
| Casp8 | TSG | 9q31 | 57.4 | 2 (10%) | 0 (0%) | 1 (4%) | 0 (0%) |
| Crebbp | POG/TSG | 10q12 | 11.6–11.7 | 0 (0%) | 1 (5%) | 0 (0%) | 0 (0%) |
| Axin1 | TSG | 10q12 | 15.4–15.5 | 1 (5%) | 1 (5%) | 0 (0%) | 0 (0%) |
| Ncor1 | TSG | 10q23 | 48.5–48.6 | 1 (5%) | 1 (5%) | 0 (0%) | 0 (0%) |
| Map2k4 | POG/TSG | 10q24 | 52.0 | 1 (5%) | 1 (5%) | 0 (0%) | 0 (0%) |
| Tp53 | POG/TSG | 10q24 | 56.4 | 1 (5%) | 1 (5%) | 0 (0%) | 0 (0%) |
| Nf1 | TSG | 10q35 | 65.6–65.8 | 1 (5%) | 1 (5%) | 0 (0%) | 0 (0%) |
| Erbb2 | POG | 10q31 | 87.2 | 0 (0%) | 1 (5%) | 0 (0%) | 0 (0%) |
| Cux1 | POG/TSG | 12q12 | 21.3–21.5 | 0 (0%) | 1 (5%) | 0 (0%) | 0 (0%) |
| Tbx3 | POG/TSG | 12q16 | 38.2 | 0 (0%) | 1 (5%) | 0 (0%) | 0 (0%) |
| Nf2 | TSG | 14q21 | 85.4–85.5 | 1 (5%) | 0 (0%) | 0 (0%) | 0 (0%) |
| Rb1 | TSG | 15q11 | 53.8–54.0 | 1 (5%) | 0 (0%) | 1 (4%) | 0 (0%) |
| Znf703 | POG | 16q12.3 | 69.4 | 0 (0%) | 1 (5%) | 0 (0%) | 0 (0%) |
| Fgfr1 | POG | 16q12.4 | 70.9 | 0 (0%) | 2 (10%) | 0 (0%) | 0 (0%) |
| Gata3 | POG/TSG | 17q12.3 | 80.0 | 3 (14%) | 2 (10%) | 0 (0%) | 0 (0%) |
| Apc | TSG | 18p12 | 26.7–26.8 | 3 (14%) | 0 (0%) | 0 (0%) | 0 (0%) |
| Smad4 | TSG | 18q12.2 | 70.4–70.5 | 3 (14%) | 0 (0%) | 1 (4%) | 0 (0%) |
| Prdm1 | TSG | 20q13 | 48.4–48.5 | 3 (14%) | 0 (0%) | 0 (0%) | 0 (0%) |
| Bcor | TSG | Xq12 | 22.7–22.8 | 4 (19%) | 0 (0%) | 1 (4%) | 0 (0%) |
| Atrx | TSG | Xq22 | 93.9–94.1 | 4 (19%) | 0 (0%) | 1 (4%) | 0 (0%) |

*(Continued)*

**Table 6.** (Continued)

| Gene symbol | Role in cancer[a] | Chromosome band | Location (Mb) | Tumors with copy-number change (n = 21) | | Moriyama et al. [33] (n = 28)[b] | |
|---|---|---|---|---|---|---|---|
| | | | | Loss | Gain | Loss | Gain |
| *Stag2* | TSG | Xq35 | 2.8–3.0 | 4 (19%) | 0 (0%) | 1 (4%) | 0 (0%) |

[a]POG, protooncogene;

TSG, tumor-suppressor gene.

[b]Includes γ-ray–induced (n = 10), neutron-induced (n = 8) and sporadic (n = 10) mammary carcinomas in SD rats [33];

details are provided in S1 Dataset, Sheet 4.

SD and COP on dark grey background), implying that these loss of heterozygosity events were caused by mitotic recombination or chromosome mis-segregation.

## Reduced expression of certain potential cancer-related genes in chromosomal regions with copy-number losses

The observation of frequent copy-number losses in specific chromosomal regions implies that important tumor-suppressor genes may be located within these regions. Among genes known to be related to human breast cancer [27], we previously reported the reduced expression of *Pten* (on 1q52, 0.53-fold), *Pik3r1* (on 2q12, 0.61-fold), and *Map3k1* (on 2q14, 0.75-fold) in mammary carcinomas of (SD×COP)F₁ rats [22]. Here, we investigated the expression of 24 different potentially cancer-related genes found in the chromosomal regions showing copy-number losses. Genes were selected based on functions reported in the NCBI Gene Database (https://www.ncbi.nlm.nih.gov/gene/). Quantitative PCR analysis revealed 11 putative tumor-suppressor genes that had reduced expression in radiation-induced carcinomas (Fig 4). These included *Asah2*, *Fas*, *Sfrp5* (selected from chromosome 1q52–54), *Il6st*, *Itga1* (chromosome 2q12–15), *Meis2*, *B2m*, *Mal*, *Nbl1* (chromosome 3q31–42), and *Id3* and *C1qa* (chromosome 5) (Fig 4). Genes for which expression did not change included *Ifit1* (chromosome 1q52), *Srek1*, *Cenpk*, *Ercc8*, *Gpbp1* (chromosome 2q12–14), *Cat*, *Bmf*, *Rad51*, *Tp53bp1*, *Dusp2*, *Bcl2l1* (chromosome 3q31–42), and *Runx3* (chromosome 5). *Plk2* (chromosome 2q14) was significantly upregulated. The downregulation of these 11 genes supports their potential relevance to radiation-induced mammary carcinogenesis.

## Discussion

Epidemiological studies have established that radiation exposure is a risk factor for cancer development in humans, and determining the genetic factors that interact with radiation in this context is vital to understanding individuals' responses to radiation [3]; however, screening of genetic polymorphisms related to environmental cancer risk in humans generally requires a massive sample size [7]. In this regard, animal models are useful for studying the role of gene-environment interactions in cancer susceptibility. Analyses of F₁ hybrids of cancer-susceptible and -resistant strains are also advantageous for identifying driver genes in cancer using experimental animal models, including those of radiation-induced carcinogenesis. We examined the mammary-cancer susceptibility of irradiated (SD×COP)F₁ hybrids of susceptible SD and resistant COP rats and found that they had intermediate susceptibility levels; thus, they are useful for exploring cancer-causing gene mutations. Our approach of combining analyses of copy-number variations and AIs in mammary cancer successfully identified marginally frequent (14–38%) copy-number losses in chromosome regions 1q52–54, 2q12–15,

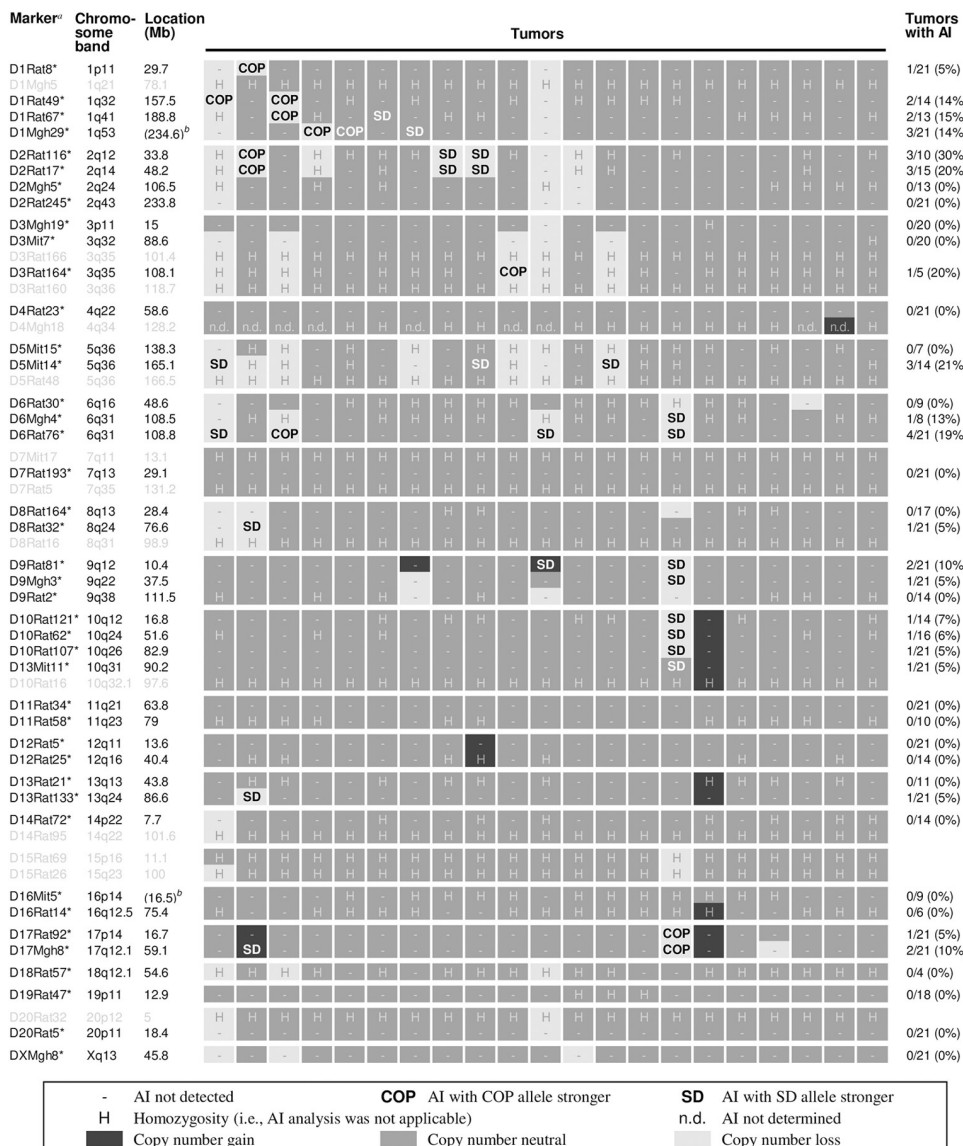

**Fig 3. Allelic imbalance (AI) sites observed in (SD×COP)F₁ mammary carcinomas.** Results of the AI analysis along with information from microarray-based copy-number analysis (Fig 2), indicated in grey scale. Columns indicate individual tumors. Note that copy-number variations irrelevant to the indicated markers are not shown. [a]Markers with (black with asterisk) or without (grey) heterozygous individuals. [b]No data present in the rn4 rat genome assembly; values in parentheses are from the Celera assembly. 'H', markers exhibiting homozygosity, where allelic analyses were impossible.

and 3q31–42 as well as chromosome 5, with many genes in these regions showing reduced expression. This frequency is higher than the AI frequency reported in radiation-induced mammary cancer of (WF×COP)F₁ rats (4–13%) [18], and higher than the frequency of copy number changes in breast cancer–relevant genes in SD rats (4–11%) [33]. Thus, (SD×COP)F₁ rats offer a new option in the search for causative genes of radiation-induced mammary cancer.

The present study is the first report of mammary carcinogenesis in irradiated COP and (SD×COP)F₁ rats. COP rats are completely resistant to chemically induced mammary

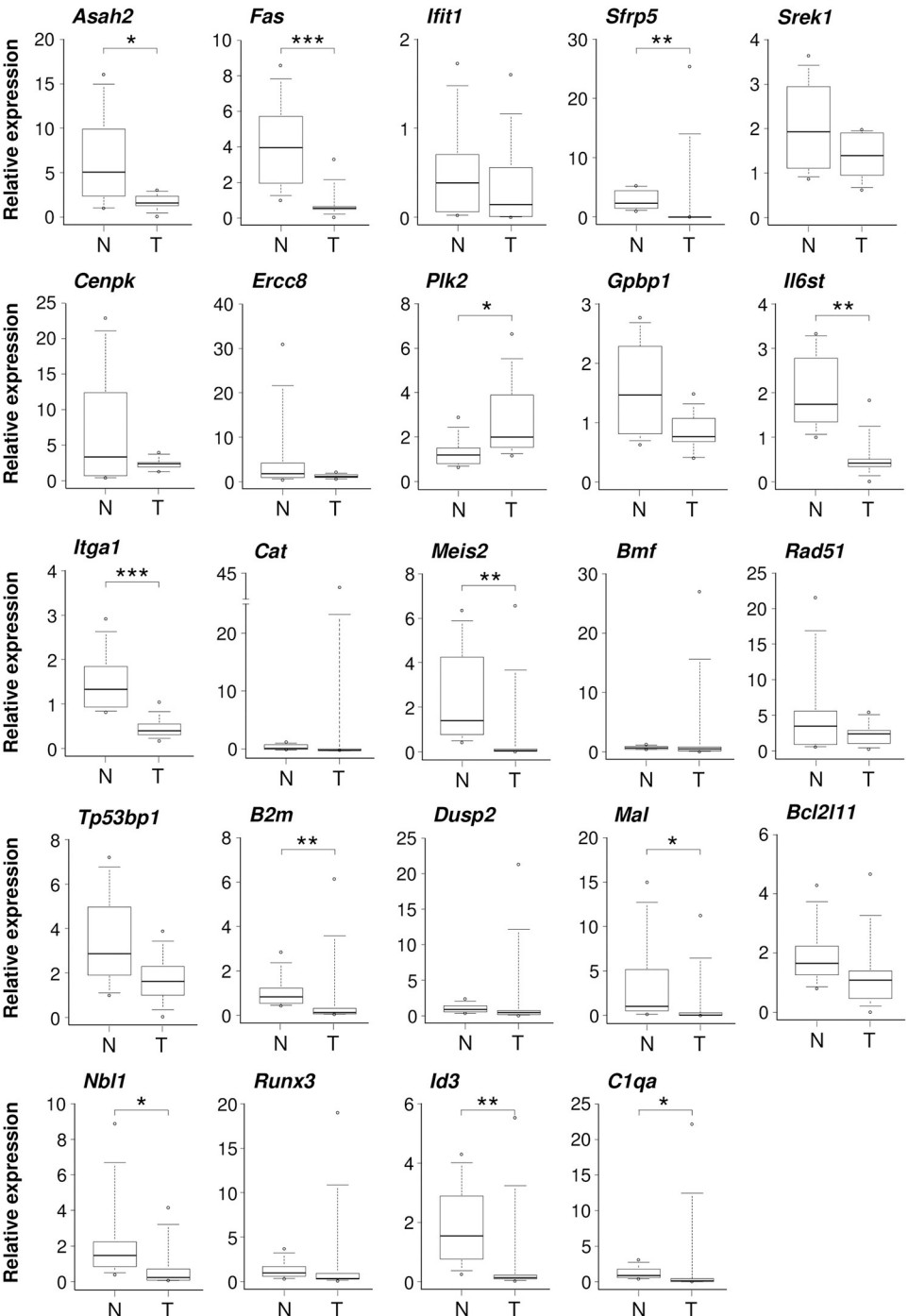

**Fig 4. Expression of 24 genes located in chromosomes 1q52–54, 2q12–15, 3q31–42, and 5 showing copy-number losses in mammary carcinomas of irradiated (SD×COP)F$_1$ rats.** Relative mRNA expression levels of the indicated genes in carcinomas and matched normal mammary glands are shown. N, normal tissues ($n$ = 7–8); T, tumors ($n$ = 10). Data are presented as box plots with median values indicated by horizontal bars within the boxes. Boxes represent values between the 25th and 75th percentiles, whiskers extend to the 5th and 95th percentiles, and circles represent outliers. $^*P < 0.05$; $^{**}P < 0.01$, $^{***}P < 0.001$ (Mann-Whitney $U$ test). Relative expression was normalized to a specific sample.

carcinogenesis [11]. In the present study, mammary carcinoma developed in irradiated COP rats after 60 weeks post-irradiation. As chemically induced tumor development usually occurs earlier (typically within 30 weeks post-induction [11]), the observation period tends to be shorter for chemically induced carcinogenesis experiments than for radiation-induced carcinogenesis experiments (e.g., 300 days in Isaacs 1988 [11] vs. >100 weeks in the present study). Thus, the lack of mammary carcinomas following chemical induction may be due to the shortened observation period.

The longer observation period in the present study also enabled evaluation of susceptibility to the development of benign mammary tumors (fibroadenoma and adenoma), which showed the same tendency as susceptibility to mammary carcinoma. This similarity might be understood given that each of these lesions is of epithelial origin and that the resistance exhibited by COP rats has been attributed to epithelial cells [34].

The inheritance of COP rat resistance to chemically induced mammary carcinogenesis has been reported to be dominant under a short observation period [11]. This is in stark contrast to our finding that the susceptibility of (SD×COP)F$_1$ hybrids was intermediate, i.e., between that of its parent strains, which is logical, as this trait is known to be polygenic [19]. The inheritance of benign mammary tumors has not yet been reported because most of the previous experiments used chemical carcinogens, which mainly induce carcinoma. The present finding indicates that inheritance of benign tumor resistance in (SD×COP)F$_1$ hybrids is similar to that of carcinoma, suggesting a common mechanism of resistance to carcinoma and benign tumors in COP rats.

The present study identifies chromosome regions 1q52–54, 2q12–15, and 3q31–42 and the entire chromosome 5 as sites of potentially relevant cancer-related genes. We previously reported aberrations of genes related to the phosphoinositide 3-kinase (PI3K) pathway, including downregulation of *Pten* (1q52) and mutations of *Pik3r1* (2q12), in mammary cancers from the same cohort of irradiated (SD×COP)F$_1$ rats [22], but the frequency of these mutations was low (i.e., 1 of 14 for each mutation). Copy number loss of *Pten* observed in (SD×COP)F$_1$ rats (24%) has also been reported in SD rats [33]. Chromosome 1q52–54 coincides with the location of the mammary-cancer susceptibility quantitative trait locus (QTL) *Mcs17*. The SD allele of this QTL is associated with an increased number of chemically induced mammary carcinomas compared with the COP allele [35]. By contrast, in our data, the polymorphic marker D1Mgh29, which is located on chromosome 1q52–54, shows a higher rate of loss of the potential susceptibility SD allele (2 of 3), suggesting a different role for this allele in the present model. Analysis of the human counterpart of this region (10q23) revealed frequent losses in sporadic breast cancers [36]. On the other hand, rat chromosome 2q14 (which harbors *Gpbp1*, *Map3k1* and *Il6st*) coincides with *Mamtr3* (also known as *Mcs1b* or *Mcs10*), another mammary-cancer susceptibility QTL for which the COP allele confers resistance to chemically induced mammary carcinogenesis [37]. *Map3k1* (43.1 Mb of chromosome 2) and *Mier3* (43.0 Mb) have been identified as candidate susceptibility genes within this region [38]. Our data indicate that loss of the markers D2Rat116 and D2Rat17, which flank *Map3k1* and *Mier3*, are biased to the potentially resistant COP allele. Analysis in humans indicated that *MAP3K1* is the gene in the corresponding region (human chromosome 5q11.2) with the greatest influence on risk of breast-cancer development [39]. No QTL has been reported on chromosome 3q31–42, suggesting the existence of important unidentified determinants of susceptibility. A further search for candidate causative genes in this region is thus warranted, and downregulated genes identified in the current study, namely *Meis2*, *B2m*, *Mal*, and *Nbl1*, are candidates for further study. Comparison with previous measurements of gene expression changes in radiation-induced SD rat mammary carcinoma [24] (S2 Fig) supports downregulation of *Fas*, *Sfrp5* (chromosome 1q52–54), *Itga1* (chromosome 2q12–15), *Meis2*, *B2m*, *Mal* (chromosome 3q31–

42), *Id3*, *C1qa*, and *Nbl1* (chromosome 5), reinforcing the role of these genes as potential tumor suppressors. A previous study on irradiated (WF×COP)F$_1$ rats revealed somewhat less frequent AI of markers located on 1q32–56 (1–3 of 24 tumors, 4–12%) and 2q24–34 (1–2 of 25 tumors, 4–8%) [18], implying that the mechanisms underlying mammary carcinogenesis differ between SD and WF rats.

Our present analysis indicated a positive correlation between large deletions and age at tumor detection, implying another benefit of the long observation period. This observation suggests that some tumors develop via accumulation of large chromosomal deletions which requires time, whereas those that develop early may use other mechanisms, such as point mutations, inversions, and translocations.

Beside the above-mentioned susceptibility loci, several factors should be considered that may have caused the observed strain difference. Burden of mammary tumor virus has been related to mammary tumor susceptibility in some mouse strains [40], and a counterpart tumor virus has been reported in rats [41], but has not been confirmed extensively; however, we cannot completely rule out the possibility that the strain difference observed in the current study is due to virus burden. Genetic contamination has been reported as another factor influencing disease susceptibility in specific rat strains [42, 43]. The current SD and COP strains were maintained under strict management; nevertheless, as genetic tests were not conducted, the possibility of contamination cannot be excluded. It is also possible that systemic factors affect tumor development. Long-term hormone administration to COP rats has been reported to promote development of mammary tumors that would otherwise undergo spontaneous regression [44]. Fat tissue is a source of estrogen, especially after senescence of ovarian function; thus, obesity and being overweight promote breast cancer development [45]. The difference in body weight among rat strains reported herein, which is concordant with that associated with mammary cancer susceptibility, is therefore a plausible factor that could explain the observed susceptibility.

In humans, sporadic breast cancers generally exhibit multiple DNA copy number aberrations, consistent with the present animal model. By contrast, there is little evidence of genetic aberrations in radiation-induced breast cancer in humans. In the present study, the regions showing copy number aberrations in tumors from (SD×COP)F$_1$ rats did not necessarily correspond to those previously reported in SD rats [33]. This strain difference suggests that copy number aberrations in radiation-induced breast cancer are subject to the influence of genetic background. In fact, in a study of human breast cancers that developed as second primary cancers after radiotherapy ($n$ = 3 tumors), no common deletions or inversions that could be causative were reported; rather, multiple deletions and inversions of non-coding regions were reported [46]. Larger investigations are thus warranted to clarify the common genetic changes detected in radiation-induced human cancers.

The present study has the following limitations. First, results should be compared between irradiated and non-irradiated groups to determine extent to which the present findings can be attributed to radiation exposure or genetic characteristics. Second, our results may have been affected by variation in exposure conditions, such as radiation dose, dose rate, fractionation regimen, radiation type, age at time of exposure, hormonal conditions at time of exposure, and whether the whole or partial body was irradiated. This point remains open for future study. Third, it is unclear whether the mammary carcinoma subtype is strain-dependent. Our previous studies on mammary carcinomas of (SD×COP)F$_1$ ($n$ = 24 tumors) and SD ($n$ = 85 tumors) rats indicated that they are mainly (70%–90%) of the luminal subtype [22, 32, 33], consistent with reports from other laboratories [47, 48]. Further studies are required to clarify if this applies to COP rats, as the number of tumors obtained in our study was small ($n$ = 9) due to the high resistance of the strain.

Taken together, the present study clearly indicates that irradiated (SD×COP)F$_1$ hybrid rats are intermediately susceptible to mammary carcinogenesis, and hence are a useful model for exploring potentially causative gene mutations in mammary cancer. Chromosome regions 1q52–54, 2q12–15, and 3q31–42 and chromosome 5 are expected to harbor driver mutations relevant to mammary carcinogenesis. This study also suggests that mammary cancer in (SD×COP)F$_1$ rats involves many genetic aberrations that are relevant to human breast cancer and thus offers a good model for basic research.

## Supporting information

**S1 Dataset. Unprocessed data.** Sheet 1: Animal and tumor development data used for Fig 1 and Tables 3–5. Sheet 2: Tumor location and tumor type data in Table 2 and age at first tumor detection data used for Table 3. Sheet 3: Quantitative PCR data used for Fig 4. Sheet 4: Copy-number aberration data used for Table 6. Sheet 5: Previous expression microarray data mentioned in Discussion.
(XLSX)

**S1 Fig. Effect of selection of tumors.** Distribution of tumor weight (A), age at tumor detection (B), age at autopsy (C), and tumor age (i.e., interval between tumor detection and autopsy) (D). Circles, individual tumors; horizontal and vertical bars, mean and SD. *P* values, Welch's *t* test.
(TIF)

**S2 Fig. Expression of genes in SD rat mammary carcinomas.** Genes in Fig 4 in mammary carcinomas from SD rats from a previous microarray analysis [24]. Expression levels are standardized against the 75th percentiles of all genes on individual microarrays and are expressed as log$_2$ values. Probe IDs are shown above the gene symbols. Data are presented as the mean and SD. * *P* < 0.05, Mann-Whitney *U* test.
(TIF)

## Acknowledgments

We thank Harumi Osada and Masami Ootawara for animal breeding and specimen preparation as well as the staff at the Laboratory Animal and Genome Sciences Section for animal facility management. A portion of this work was conducted in association with the QST-NIRS project of the Japan StoreHouse of Animal Radiobiology Experiments (J-SHARE) [49].

## Author Contributions

**Conceptualization:** Kazumasa Inoue, Masahiro Fukushi, Shizuko Kakinuma, Yoshiya Shimada.

**Formal analysis:** Kazuhiro Daino, Atsuko Ishikawa, Tatsuhiko Imaoka.

**Funding acquisition:** Kazuhiro Daino, Shizuko Kakinuma, Tatsuhiko Imaoka.

**Investigation:** Mayumi Nishimura, Kazuhiro Daino, Maki Fukuda, Ikuya Tanaka, Hitomi Moriyama, Kaye Showler, Yukiko Nishimura, Masaru Takabatake, Toshiaki Kokubo, Tatsuhiko Imaoka.

**Project administration:** Shizuko Kakinuma, Yoshiya Shimada.

**Software:** Atsuko Ishikawa.

**Supervision:** Kazumasa Inoue, Masahiro Fukushi, Yoshiya Shimada.

**Writing – original draft:** Mayumi Nishimura, Kazuhiro Daino, Tatsuhiko Imaoka, Yoshiya Shimada.

**Writing – review & editing:** Maki Fukuda, Ikuya Tanaka, Hitomi Moriyama, Kaye Showler, Yukiko Nishimura, Masaru Takabatake, Toshiaki Kokubo, Atsuko Ishikawa, Kazumasa Inoue, Masahiro Fukushi, Shizuko Kakinuma.

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
