## [Decision Letter · Decision Letter 0]

15 May 2021

PONE-D-21-05276

Radiation induces mammary cancer in F1 hybrids of susceptible Sprague-Dawley and resistant Copenhagen rats, with copy-number losses that pinpoint potential tumor suppressors

PLOS ONE

Dear Dr. Imaoka,

Thank you for submitting your manuscript to PLOS ONE. After careful consideration, we feel that it has merit but does not fully meet PLOS ONE’s publication criteria as it currently stands. Therefore, we invite you to submit a revised version of the manuscript that addresses the points raised during the review process.

We look forward to receiving your revised manuscript.

Kind regards,

Soile Tapio

Academic Editor

PLOS ONE

Journal Requirements:

[This work was supported in part by the Japan Society for the Promotion of Science (https://www.jsps.go.jp/english/) via JP17K00562 (KD) and JP20K12176 (KD). The funders had no role in study design, data collection and analysis, decision to publish, or preparation of the manuscript.].    

We note that one or more of the authors are employed by a commercial company: Konica Minolta Inc.

3. Please amend your authorship list in your manuscript file to include author Masaru Takabatake.

Reviewers' comments:

Reviewer's Responses to Questions

**Comments to the Author**

1. Is the manuscript technically sound, and do the data support the conclusions?

Reviewer #1: Yes

Reviewer #2: Partly

2. Has the statistical analysis been performed appropriately and rigorously? 

Reviewer #1: Yes

Reviewer #2: Yes

3. Have the authors made all data underlying the findings in their manuscript fully available?

Reviewer #1: Yes

Reviewer #2: Yes

4. Is the manuscript presented in an intelligible fashion and written in standard English?

Reviewer #1: Yes

Reviewer #2: No

5. Review Comments to the Author

Reviewer #1: This manuscript described the genetic analysis of radiation-induced cancer induced in parental (SD and COP) as well as F1 rats. The use of susceptible and resistant rat strains in radiation carcinogenesis studies is not itself novel, and some features of this work have already been published, The molecular analysis of the genetic changes occurring in the F1 tumors has merit as an original observation. However, parts of the study are not clearly documented and there appears to be no consideration given to the fact that most, if not all, of the alterations may be due to genetic noise and/or can occur equally in sporadic mammary cancers.

The work comes from a laboratory internationally recognized for the quality of the animal research and all appropriate animal protection measures are followed. The paper is very clearly written and the language is excellent.

The results are of broad relevance in the fields of breast cancer and in radiation-induced cancer, and therefore deserving of publication with some major and some minor changes needed to the manuscript (see below). Once these comments have been addressed or rebutted the manuscript is acceptable for publication.

Comments to be considered in a revised manuscript.

Major revision required:

I) Variability in genetic purity, animal health, housing and treatment are all recognized as confounding factors in lifespan studies. Although the authors clearly state the study is based on historical data, it is not clear to the reviewer when the study of the different animals was done. Were the SD, COP and F1 rats all irradiated and observed at the same time, or were the studies done sequentially?

II) I was not able to understand the claims made for the relevance of the study results to radiation-induced versus sporadic cancer. I was unable to determine the sporadic cancer incidence in the three animal groups, nor was it possible to determine which genetic events are present in sporadic versus radiation-induced cancers.

III) The KM plots in Figure 1 do not show the claimed similarity between the incidence of MaCa in F1 and SD animals. The plot appears to this reviewer to show an intermediate phenotype suggestive of multiple alleles influencing susceptibility. Please either show statistical evidence or revise the text to remove suggestions that imply that the SD genotype has a dominant suppressive effect on radiation sensitivity.

IV) The assignment of causality to genetic alterations found in a tumor can be misleading. In the final sentence of the introduction the authors imply the changes they observe may be causal. This is not a valid argument, rather the changes shown can only be termed as being associated with the development of the cancer. If they are passengers, causal or enabling mutations remains unproven.

V) The possible strain-specific differences in tumor type (Adenocarcinoma, fibroadenoma or luminal/non-luminal) and location have not been discussed. The histopathological analysis of the tumors is missing. (See also point iii).

VI) The different sensitivities of the two strains, and the F1, are probably rather specific to the dose of radiation and age at irradiation, with each strain having different tumor growth rates that influence the time of detection. The factors governing the decision to use a single dose of 4Gy and the selection for the age at irradiation should therefore be explained (I assume coherence with previous literature studies?).

VII) Gene expression in normal tissue is used as control for expression in cancers. This was done on a subset of tumors. How were these tumors selected and what type of tumors were considered? The influence of size and location, as well as tumor type should be considered. Moreover, the use of normal mammary tissue includes of course considerable cell heterogeneity, so it is important to describe how normal tissue was selected and prepared.

VIII) The genetic findings in the recent study of Moriayama, including the senior author of this paper using N- irradiated SD rats (Moriyama 2021 DOI: https://doi.org/10.21873/anticanres.14751 should be compared in much greater detail to indicate differences..

IX) The tumor incidence data is censored, but it is not clear what exclusion criteria were used, nor if the F1 and COP had different competing causes.

X) The very important data presented in Figure 4 is drawn from a subset of the tumors available,. How was the selection made and most importantly how was the quantification of expression mathematically performed. A comment on the change of expression of these genes seen in COP or SD tumors would greatly improve the manuscript.

Minor points that should be considered

i) Animal numbers used are hard to find in the present manuscript, it would greatly improve readability if the n= convention was used in the text.

ii) The data in Table 4 show human genes mapping to regions of CNV/ in the rat tumors needs clarification. How were these homologies mapped?

iii) The paper would have been greatly improved by inclusion of discussion on alternative causes of the different susceptibilities e.g. mammary virus tumor burden, strain genetic contamination, animal weight and lifespan, speed of tumor growth in the two strains, tumor multiplicity.

iv) Abstract lines 45-46. This is confusingly written. If the incidence in F1 animals is intermediate it cannot be similar to SD. If it were similar to SD then the “sensitivity” genes in SD are dominant over the “resistant” genes in COP.

v) Benign mammary tumors ? what are these and how were they defined?

vi) Line 80. The “combination” of SD and COP rats …. I think this is not a clear term, it probably refers to studies on F1 hybrids?

vii) Line 84-87 The authors need to clarify if the AI on Chr1 is due to gain or loss.

. viii) What do the authors mean by saying some of the CNV are allelic imbalances?

ix) The text on lines 131-133 is trivial and does not need to be included.

x) The criteria were used to select and mark the loci selected in Figure 2 in blue are not given in enough detail and appear to be restricted to the largest chromosomes. For example changes around the Nbl1 locus are only seen in 1 tumor.

xi) The data shown in figure 3 is not clearly described. How was the assignment of gain/loss made, and how many tumors were showing the indicated changes. What is H in the individual boxes signifying?

Reviewer #2:

The manuscript submitted by Nishimura and Co is a study designed to investigate the role of genomic susceptibility in radiation-induced cancer.

The study deals with an important issue after radiotherapy. The experiment was designed and performed well, but the manuscript didn't report and interpret data well enough. The first issue is that authors need to rearrange (shape) the introduction and discussion in the focus of readers of PLOS ONE.

The authors need to start both sections with issues about radiation-induced cancer, epidemiological knowledge, and the topics like genomic susceptibility, individual sensitivity, age and gender in the context of available knowledge about radiation-induced cancer. They can of course argue the lack of suitable in vivo or in vitro model in the field and promote their model.

Otherwise, the paper remains at a level of a technical report to introduce an animal model for such a study. If this is the aim of the paper, then they need to negotiate with the editor to submit the paper as a technical report.

Even in a form of technical paper, authors still need to justify the selected criteria such as the age of animals, the dose and the dose rate, the time after exposure.

Is there any available data that a higher or lower dose and a long(er) time after exposure can affect the gene profile analysed in the present study differently?

The authors managed to provide evidence that distinguished their model from previous studies, they need to highlight those such as multiple copies no variation and try to translate it to a human scenario.

As mentioned before, the obtained data need to be discussed in the context of radiation-induced cancer.

6. PLOS authors have the option to publish the peer review history of their article (what does this mean?). If published, this will include your full peer review and any attached files.

Reviewer #1: **Yes: **Michael J Atkinson

Reviewer #2: No

---

## [Author Response · Author response to Decision Letter 0]

16 Jul 2021

Please refer to the attached letter, in which we have responded to all comments, since our responses include some figures and tables.

---

## [Decision Letter · Decision Letter 1]

28 Jul 2021

Development of mammary cancer in γ-irradiated F1 hybrids of susceptible Sprague-Dawley and resistant Copenhagen rats, with copy-number losses that pinpoint potential tumor suppressors

PONE-D-21-05276R1

Dear Dr. Imaoka,

We’re pleased to inform you that your manuscript has been judged scientifically suitable for publication and will be formally accepted for publication once it meets all outstanding technical requirements.

Kind regards,

Soile Tapio

Academic Editor

PLOS ONE

Additional Editor Comments (optional):

Reviewers' comments:

Reviewer's Responses to Questions

**Comments to the Author**

1. If the authors have adequately addressed your comments raised in a previous round of review and you feel that this manuscript is now acceptable for publication, you may indicate that here to bypass the “Comments to the Author” section, enter your conflict of interest statement in the “Confidential to Editor” section, and submit your "Accept" recommendation.

Reviewer #1: All comments have been addressed

Reviewer #2: All comments have been addressed

2. Is the manuscript technically sound, and do the data support the conclusions?

Reviewer #1: Yes

Reviewer #2: Yes

3. Has the statistical analysis been performed appropriately and rigorously? 

Reviewer #1: Yes

Reviewer #2: N/A

4. Have the authors made all data underlying the findings in their manuscript fully available?

Reviewer #1: Yes

Reviewer #2: Yes

5. Is the manuscript presented in an intelligible fashion and written in standard English?

Reviewer #1: Yes

Reviewer #2: Yes

6. Review Comments to the Author

Reviewer #1: THank you for addressing all of my comments in a constructive and scientifically valid manner. The manuscript is now acceptable for publication.

Reviewer #2: Dear Editor,

The authors have addressed all my concerns and issues well. The manuscript should now qualify for publication in PlosOne.

best regards

7. PLOS authors have the option to publish the peer review history of their article (what does this mean?). If published, this will include your full peer review and any attached files.

Reviewer #1: **Yes: **Michael J. Atkinson

Reviewer #2: No

---

## [Editor Report · Acceptance letter]

5 Aug 2021

PONE-D-21-05276R1 

Development of mammary cancer in γ-irradiated F_1_ hybrids of susceptible Sprague-Dawley and resistant Copenhagen rats, with copy-number losses that pinpoint potential tumor suppressors 

Dear Dr. Imaoka:

I'm pleased to inform you that your manuscript has been deemed suitable for publication in PLOS ONE. Congratulations! Your manuscript is now with our production department. 

Kind regards, 

on behalf of

Dr. Soile Tapio 

Academic Editor

PLOS ONE